

# Climate change signal in the ocean circulation of the Tyrrhenian Sea

Alba de la Vara[1,2]*, Iván M. Parras-Berrocal[3], Alfredo Izquierdo[3], Dmitry V. Sein[4,5], William Cabos[6]

[1] Environmental Sciences Institute, University of Castilla-La Mancha, Avenida Carlos III s/n, 45071, Toledo, Spain

[2] Departamento de Matemática Aplicada a la Ingeniería Industrial, E.T.S.I. Industriales, Universidad Politécnica de Madrid, c/ José Gutiérrez Abascal, 2, 28006 Madrid, Spain

[3] Department of Applied Physics, Faculty of Marine and Environmental Sciences, Marine Research Institute (INMAR), International Campus of Excellence of the Sea (CEI·MAR), University of Cadiz, Puerto Real, 11510 Cadiz, Spain

[4] Alfred Wegener Institute for Polar and Marine Research, Bremerhaven, Germany

[5] P. P. Shirshov Institute of Oceanology RAS, Moscow, Russia

[6] Departmento de Física y Matemáticas, Universidad de Alcalá, Madrid, 28801, Spain

*Correspondence to*: Alba de la Vara (adelavaraf@gmail.com)

**Abstract.** The Tyrrhenian Sea plays an important role in the winter deep water formation in the North Western Mediterranean through the water that enters the Ligurian Sea via the Corsica Channel. Therefore, the study of the impact of

the changes in the future climate on the Tyrrhenian circulation and its consequences represents an important issue. Furthermore, the seasonally-dependent, rich in dynamical mesoscale structures, Tyrrhenian circulation is dominated by the interplay of local climate and the basin-wide Mediterranean circulation via the water transport across its major straits and an adequate representation of its features represents an important modeling challenge. In this work we examine with a regionally-coupled atmosphere-ocean model the changes in the Tyrrhenian circulation by the end of the 21st century under

the RCP8.5 emission scenario, their driving mechanisms, as well as their possible impact on winter convection in the NW Mediterranean. Our model successfully reproduces the main features of the Mediterranean Sea and Tyrrhenian basin present-day circulation. We find that toward the end of the century the winter cyclonic, along-slope stream around the Tyrrhenian basin becomes weaker. This weakening increases the wind work transferred to the mesoscale structures, which become more intense than at present in winter and summer. We also find that, in the future, the northward water transport across the

Corsica Channel towards the Liguro-Provençal basin becomes smaller than today. Also, water that flows through this channel presents a stronger stratification because of a generalized warming with a saltening of intermediate waters. Both factors may contribute to the interruption of deep water formation in the Gulf of Lions by the end of the century.

## 1. Introduction

30       The Mediterranean Sea is a semi-enclosed basin with an only connection to the open ocean via the narrow and shallow Strait of Gibraltar (∼14 km width, ∼300 m depth). It is connected to the Black Sea through the Marmara-Bosphorus system and is divided into the Eastern and Western Mediterranean by the Strait of Sicily. Despite its small size, the Mediterranean Sea presents a thermohaline circulation which is driven by negative surface heat and freshwater budgets i.e., it experiences a net heat and water loss to the atmosphere (e.g., Jordà et al., 2017), with deep and intermediate-depth open convection at



specific sites in response to winter cooling (Bergamasco and Malanotte-Rizzoli, 2010). The surface component of the thermohaline circulation, the Atlantic Waters (AWs) flow into the Mediterranean Sea through the Strait of Gibraltar and then reach the Eastern Mediterranean via the Sicily Strait. On its way to the east, the AWs experience a surface buoyancy loss due to evaporation. In the Levantine basin, water and heat loss due to air-sea fluxes drives intermediate-depth convection (down to 150–350 m) forming the so-called Levantine Intermediate Water, LIW (Wüst, 1961; Lascaratos et al., 1993; LIWEX

group, 2003). This salty water mass, the LIW, flows to the west and then to the Atlantic Ocean, along with other Mediterranean intermediate and deep waters (Millot 2009, 2019) through the deepest portion of the Strait of Gibraltar. In turn, the increased salt content of the LIW preconditions surface waters for deep water formation in winter in locations such as the Gulf of Lions, the Adriatic or the Aegean (MEDOC Group, 1970).

In the Western Mediterranean, the Tyrrhenian Sea is well known for its complex configuration and bathymetry.

Furthermore, its basin-scale circulation presents a strong seasonal variability and a very active, also seasonal-dependent, mesoscale activity which are set by the interaction of the water exchange through the Strait of Sicily, the Corsica Channel and the Strait of Sardinia (see their location in Fig. 1), as well as by local atmospheric fluxes (e.g., Krivosheya and Ovchinnikov, 1973; Korres et al., 2000). The Tyrrhenian water column is composed by Modified Atlantic Water (MAW) at the surface, LIW at intermediate depths and Tyrrhenian Deep Water (TDW) at deeper levels (Astraldi et al., 2002; Vetrano et

al., 2004). Observations and modeling works indicate a cyclonic winter circulation occupying the whole basin starting from a stream of MAW that enters the Tyrrhenian through the Sardinia Strait (Krivosheya and Ovchinnikov, 1973; Artale et al., 1994; Zavatarelli and Mellor, 1995), which is interrupted in summer (Artale et al., 1994; Gasparini et al., 2008). Below MAW, the LIW crosses the Sicily Strait and gets into the Tyrrhenian Sea, where flows affected by the cyclonic circulation. At the Corsica channel, part of LIW deflects southward along the eastern coast of Corsica and Sardinia, while the rest

continues flowing northward to the North Western Mediterranean forming the so-called Ligurian current (Sciascia et al., 2019). Thus, the outflow of LIW through the Corsica Channel to the Liguro-Provençal region is important as it may directly impact the so-called preconditioning phase (MEDOC Group, 1970) of deep water formation in the Gulf of Lions (Schroeder et al., 2010). Regarding the mesoscale circulation, permanent and recurrent structures have been spotted over the last years thanks to the use of satellite images and high-resolution ocean circulation models (Béranger et al., 2004; Vetrano et al.,

2010; Iacono et al., 2013; Napolitano et al., 2016; de la Vara et al., 2019). In particular, mesoscale circulation has been found to be especially active in summer.

In the current context of climate change, the Mediterranean Sea has been identified as one of the regions in which the warming signal is amplified. Thus, this sea is widely considered as a climate change "hot-spot" for the scientific community (Giorgi, 2006). This projected warming may have an impact on the air-sea fluxes, deep water convection (Parras-Berrocal et

al., 2020) and in the overall Mediterranean Sea circulation (Soto-Navarro et al., 2020). As stressed above, the Tyrrhenian Sea shows seasonal variability and features recurrent structures that are heavily dependent on the large-scale Mediterranean circulation and the local air-sea fluxes. Therefore, the impact that the changes induced by the future warming in the ocean and the atmosphere may have on the circulation in that region is of particular interest. Also, waters that exit the Tyrrhenian via the Corsica Channel have an impact on the hydrology of the Liguro-Provençal area and thus in the deep water formation





in the Gulf of Lions (Astraldi et al., 1999). As climate change simulations with state-of-the-art global models do not have the necessary horizontal resolution, the best tool available for the study of changes in the regional ocean circulation and the corresponding driving mechanisms are the Atmosphere-Ocean Regionally Coupled Models (AORCMs; e.g., Rummukainen 2016). Only the oceanic and atmospheric components of regional models can reach the necessary horizontal resolution to simultaneously capture basin-scale and mesoscale processes in this region (e.g., Somot et al., 2006). The main aim of this

work is to study the impact of climate change on the variability of the surface circulation of the Tyrrhenian Sea by the end of the century, as well as to identify the implied driving mechanisms and to examine the possible impact of these changes on winter convection in the NW Mediterranean Sea. To that end we use the regionally-coupled climate model ROM (Sein et al. 2015), which has been extensively adopted to study aspects related to the climate of the Mediterranean Sea (e.g., Darmaraki et al., 2019; Parras-Berrocal et al., 2020). This article is structured as follows. In Section 2, the model setup is described. In

Section 3, the present-day and future geostrophic circulation of the Tyrrhenian basin and changes in the transport across the Sardinia Strait and the Corsica Channel are studied. In Section 4, the causes and the possible consequences of the reported changes are analyzed. The Conclusions are offered in Section 5.

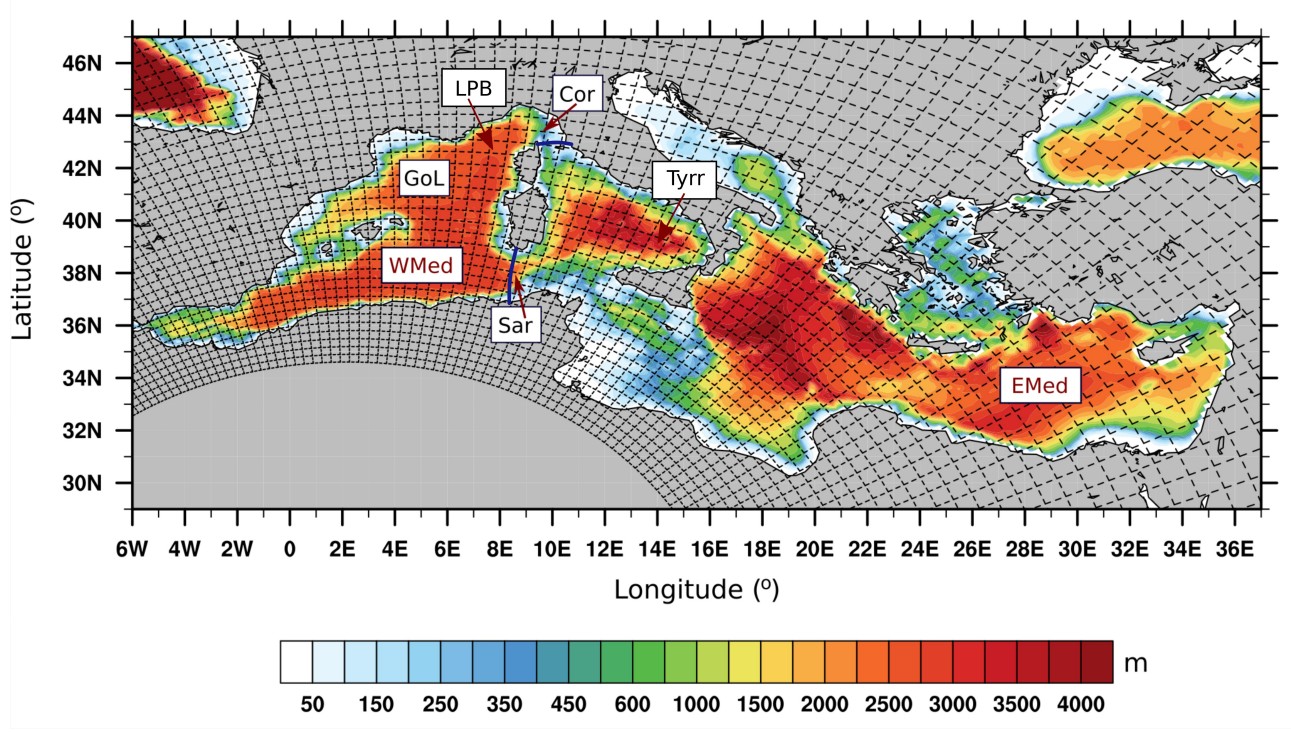

**Figure 1: Bathymetry of the Mediterranean Sea adopted in ROM, in m, together with the oceanic grid (only 1 out of 4 lines are drawn). "WMed" and "EMed" refer to the Western and Eastern Mediterranean Sea, respectively. The**
**arrows next to "Sar" and "Cor" indicate the location of the transects in which the transport across the Sardinia Strait and the Corsica Channel are measured (their coordinates are indicated in Figs. 4 and 5 of the manuscript). "Tyrr" stands for Tyrrhenian, "GoL" for Gulf of Lions and "LPB" to Liguro-Provençal basin.**

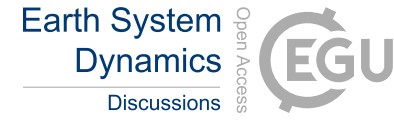

## 2. Model setup

Here we use the regionally-coupled climate model ROM (REMO-OASIS-MPIOM; Sein et al., 2015) comprising the atmospheric component REMO (REgional MOdel; Jacob and Podzun 1997; Jacob 2001) the oceanic component MPIOM (Max Planck Institute Ocean Model; Maier-Reimer 1997; Marsland et al., 2003; Jungclaus et al., 2013) and the OASIS coupler (Valcke et al., 2003). REMO (Déqué et al., 2012; Jimenez-Guerrero et al., 2013; Kotlarski et al., 2014) and ROM (Sein et al., 2014; Cabos et al., 2018; Darmaraki et al., 2019) have been successfully applied to study the regional climate

and climate change signal in different regions of the world, with a performance comparable to the best state-of-the-art models (Cabos et al., 2017; Sein et al., 2020).

MPIOM is a well-established global ocean model that has not only been extensively used for global climate studies as part of the different versions of MPI-ESM (Max-Planck-Institut für Meteorologie Earth System Model; Giorgetta et al., 2013), but also for regional studies (e.g. Mathis et al., 2018; Izquierdo and Mikolajewicz 2019; Liu et al., in press). MPIOM

has an orthogonal curvilinear grid that, in this configuration, achieves a maximum horizontal resolution of about 7 km (mesoscale eddy-rich) near the Gibraltar Strait, decreasing down to 25 km at the eastern coast of the Mediterranean. On the vertical, MPIOM counts with 40 unevenly spaced $z$-levels with a variable thickness that ranges from 16 m at the surface to 650 m near the seafloor. Unlike most of the regional models devoted to the study of the Mediterranean Sea, the water exchange at Gibraltar and the Dardanelles in ROM is not parameterized and Atlantic water properties are not relaxed

towards climatological values in the areas adjacent to the straits.

In REMO a rotated grid with the center of the model domain located around the equator is used. It has a constant horizontal resolution of 0.25º and a total of 27 hybrid levels. The atmospheric domain includes both the North Atlantic and the Mediterranean Sea (see Fig. 1S). The global Hydrological Discharge model (HD, Hagemann and Dümenil Gates, 2001) computes river runoff at 0.5º resolution and is coupled to both the ocean and the atmosphere.

OASIS (Valcke et al., 2003) is used to couple REMO and MPIOM and atmospheric and oceanic fields are exchanged with a 3-hour coupling time step (see Sein et al., 2015). For more details on the ROM setup we refer the reader to Cabos et al. (2020). ROM has been shown to be able to provide a regional signal that is significantly different from the one provided by the driving model (Limareva et al., 2017; Cabos et al., 2018; Sein et al., 2020).

To study the impact of climate change on the mesoscale circulation of the Tyrrhenian, we use a simulation in which

ROM is forced by the low-resolution version of MPI-ESM under the high emission RCP8.5 CMIP5 scenario (Taylor et al., 2012), which is expected to have the strongest impact on the Tyrrhenian circulation. ROM and MPI-ESM share the same oceanic component, but the MPIOM setup used in MPI-ESM differs from the setup used in ROM. In particular, MPI-ESM has the higher horizontal resolution in the North Atlantic deep water formation areas, with a minimum grid spacing of about 15 km around Greenland (Jungclaus et al., 2006). In ROM, the higher resolution areas are located near the North America

and North Africa coasts, having the highest resolution in the southern western Mediterranean and Northwest Africa, reaching 7 km in the Gibraltar Strait and the region of the Alboran Sea. Also, ROM has a good representation of the diurnal cycle (3



hours coupled time step), which is neglected in the MPI-ESM simulations. The climate change simulations start from the last state of the present-time simulations and cover the 2006-2099 time period.

## 3. Results

In this section we focus on the geostrophic circulation in order to use available gridded products derived from satellite altimetry for validation. First we show the present-day (1976-2005) Mediterranean-scale geostrophic circulation simulated by ROM. Then, we concentrate on the validation of the present-day geostrophic circulation of the Tyrrhenian Sea and study how its basin-scale and mesoscale-size circulation changes by the end of the century (2070-2099). Additionally, we also examine present-day and future vertical sections of velocity, salinity and temperature at the Sardinia Strait and the

Corsica Channel.

### 3.1. Present-day Mediterranean geostrophic circulation

    Here we study the ability of ROM to represent the seasonal, present-day geostrophic circulation of the Mediterranean Sea in comparison to high-resolution altimeter data from AVISO (www.marine.copernicus.eu). AVISO

provides multimission altimeter gridded sea-surface heights and derived variables over the Mediterranean Sea with a horizontal resolution of 1/8° × 1/8°. The reason why we focus on the surface geostrophic circulation is twofold. On the one hand, a complete validation of the ability of ROM to reproduce the main features of the Mediterranean Sea characteristics has already been done in Parras-Berrocal et al. (2020). On the other hand, an adequate representation of the main features of the present-day circulation is required in order to study the future circulation and determine the causal mechanisms

responsible for the changes in the geostrophic circulation by the end of the 21$^{st}$ century.

    In winter ROM captures the main features of the Mediterranean geostrophic circulation inferred from AVISO (Figs. 2A and 2C). This is, AWs entering the Mediterranean Sea at the surface and the development of the so-called Alboran anticyclonic gyres. AWs keep on flowing along the African coast on their way to the east forming the Algerian current, characterized by a large mesoscale activity (Millot, 1985). Part of this AWs, modified in the Algerian basin, flow north to the

Ligurian and Provençal areas, enter the Tyrrhenian Sea and the remaining waters go through the Sicily Strait to the Eastern Mediterranean (Ovchinnikov, 1966). Then, MAW continues to flow along-slope to complete a cyclonic gyre around the Mediterranean Sea. Also there are areas are characterized by large negative values of SSH associated to the sinking of surface waters due to convection.

    In summer, ROM also reproduces well the main characteristics of the Mediterranean circulation from AVISO

(Figs. 2B and 2D). First, ROM shows the slow down in the geostrophic velocities along the African coast. Also, the summer intensification of the Alboran Gyres is properly captured (Vargas-Yáñez et al., 2002; Dastis et al., 2018). In this season, the





along-slope, cyclonic gyre around the Mediterranean remains, but most of the MAW flows via the Sicily Strait with little inflow into the Tyrrhenian Sea.

From a quantitative perspective, AVISO shows stronger mesoscale activity, larger spatial variability of the sea-surface height, SSH, and higher geostrophic velocities than ROM. Nevertheless, ROM resolution in the region allows for a realistic representation of the main features of the surface circulation, especially in the western Mediterranean (Parras-Berrocal et al., in preparation). Overall, we can conclude that ROM is able to capture the main features of the Mediterranean circulation and its seasonal variability and the results are comparable to those obtained with other well-established models (e.g., Bergamasco and Malanotte-Rizzoli, 2010).

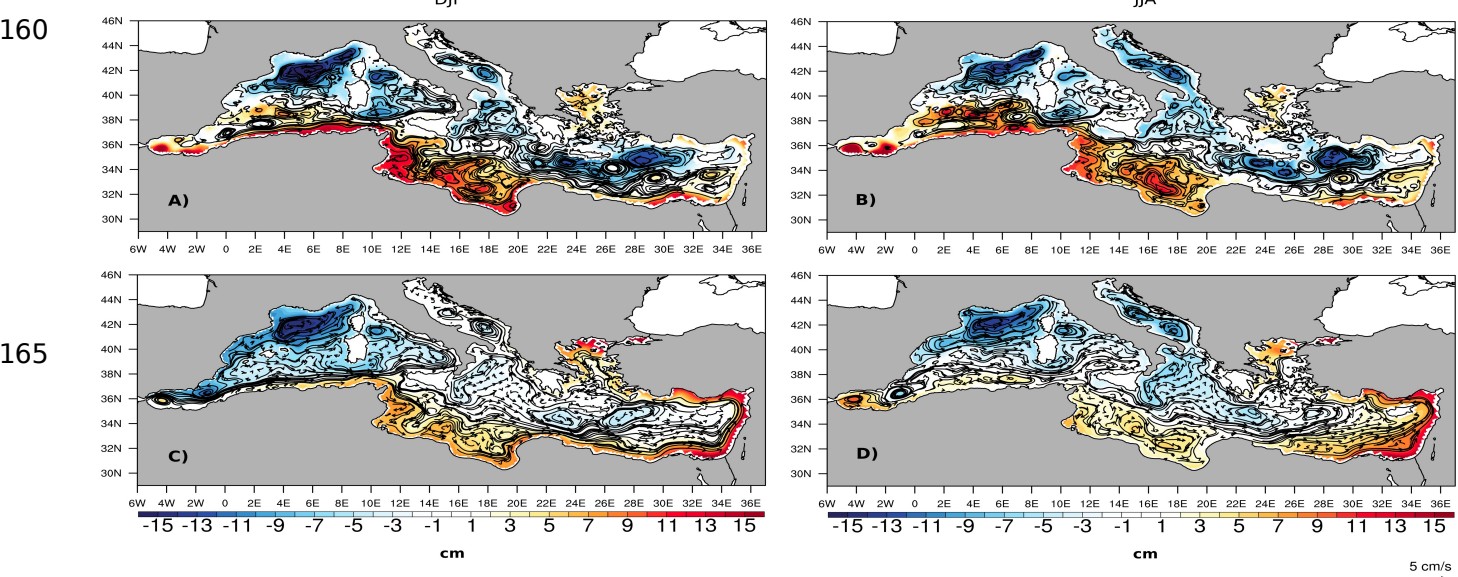

**Figure 2: Winter (left column) and summer (right column) averages of present-day geostrophic circulation (vectors, cm/s) and sea-surface height (SSH, colors, cm) from AVISO (A, B) and ROM (C, D). Note that present-day in ROM includes the 1976–2005 time period, while in AVISO this extends from 1993-2005 due to the lack of available data. Only one out of four vectors is plotted.**

## 3.2. Present-day and future geostrophic circulation in the Tyrrhenian Sea

In winter, in agreement with AVISO and previous modeling and observational works, the large-scale circulation pattern that arises from ROM points to a basin-wide, cyclonic stream around the Tyrrhenian Sea (Figs. 3A and 3B). MAW enters the Tyrrhenian through the southeastern part of the Sardinia Strait by the western tip of Sicily. Part of this flow completes the cyclonic, along-slope gyre, exiting via the northwestern portion of the Sardinia Strait, where a side-by-side counter-flow is observed, whilst part of it enters the Ligurian region via the Corsica Channel. Two prominent cyclonic, dynamical structures are overlapped to the basin-wide cyclonic gyre. The first one is the Bonifacio Gyre, which arises to the





east of this narrow strait separating Corsica and Sardinia and is related to the wind funneling through the Strait of Bonifacio (Artale et al., 1994; Vigo et al., 2019). The second one is a recirculation structure off Sardinia at the southern boundary of the Tyrrhenian Sea. These structures have been identified as permanent or quasi-permanent features of the Tyrrhenian

(Artale et al., 1994; Rio et al., 2007; Rinaldi et al., 2010; Vetrano et al., 2010; Iacono et al., 2013; Napolitano et al., 2016).

In summer, the basin-scale circulation presents important differences with respect to the winter situation and ROM is able to capture the main features of the typical summer pattern (Figs. 3D and 3E). For instance, the winter basin-wide cyclonic circulation around the Tyrrhenian does not arise. Whilst, as in winter, MAW enters the basin through the southern part of the Sardinia Strait, the development of several dynamical structures to the east of the basin reverses the circulation

along the southern and eastern coasts of the Tyrrhenian, hampering the direct connection between the southern and northern openings. In general, more dynamical structures develop than in winter. The Bonifacio Gyre stretches zonally and the water transport across the Corsica Channel becomes smaller (Astraldi and Gasparini 1992). The recirculation structure near Sardinia becomes more defined. The anticylonic circulation becomes predominant to the southeast of the Tyrrhenian, over the Sicilian coast (Iacono et al., 2013; Napolitano et al., 2016; de la Vara et al., 2019).

In the projected scenario, the winter circulation patterns are qualitatively similar to those found in the present-day simulation (Figs. 3A-3C). However, the Tyrrhenian-basin cyclonic circulation becomes weaker and the previously two mentioned dynamical mesoscale structures become larger and more intense, especially the Bonifacio Gyre. This may be related to the weakening of the basin-scale cyclonic stream, which allows for the intensification and zonal stretching of these structures. In other words, although the present-time winter circulation patterns are still visible, a trend towards the typical

present-day summer geostrophic circulation is observed i.e., a decrease in the intensity of the basin-scale current velocity pattern, larger and faster dynamical structures and a reduced water transport to the Ligurian basin through the Corsica Channel. The future summer circulation resembles a lot the current summer pattern (Figs. 3D-3F). This notwithstanding, in line with what was found for winter, a clear intensification of the dynamical structures and of the reversed eastern boundary current is observed. The zonal stretching of the Bonifacio Gyre limits, even more than at present, the water transport across

the Corsica Channel. In general, we can conclude that, although the typical winter and summer patterns can be still recognized, a transition towards a summer pattern can be observed. These results are consistent with what can be expected in a warmer climate and the implied causal mechanisms will be analyzed in detail in the Discussion.







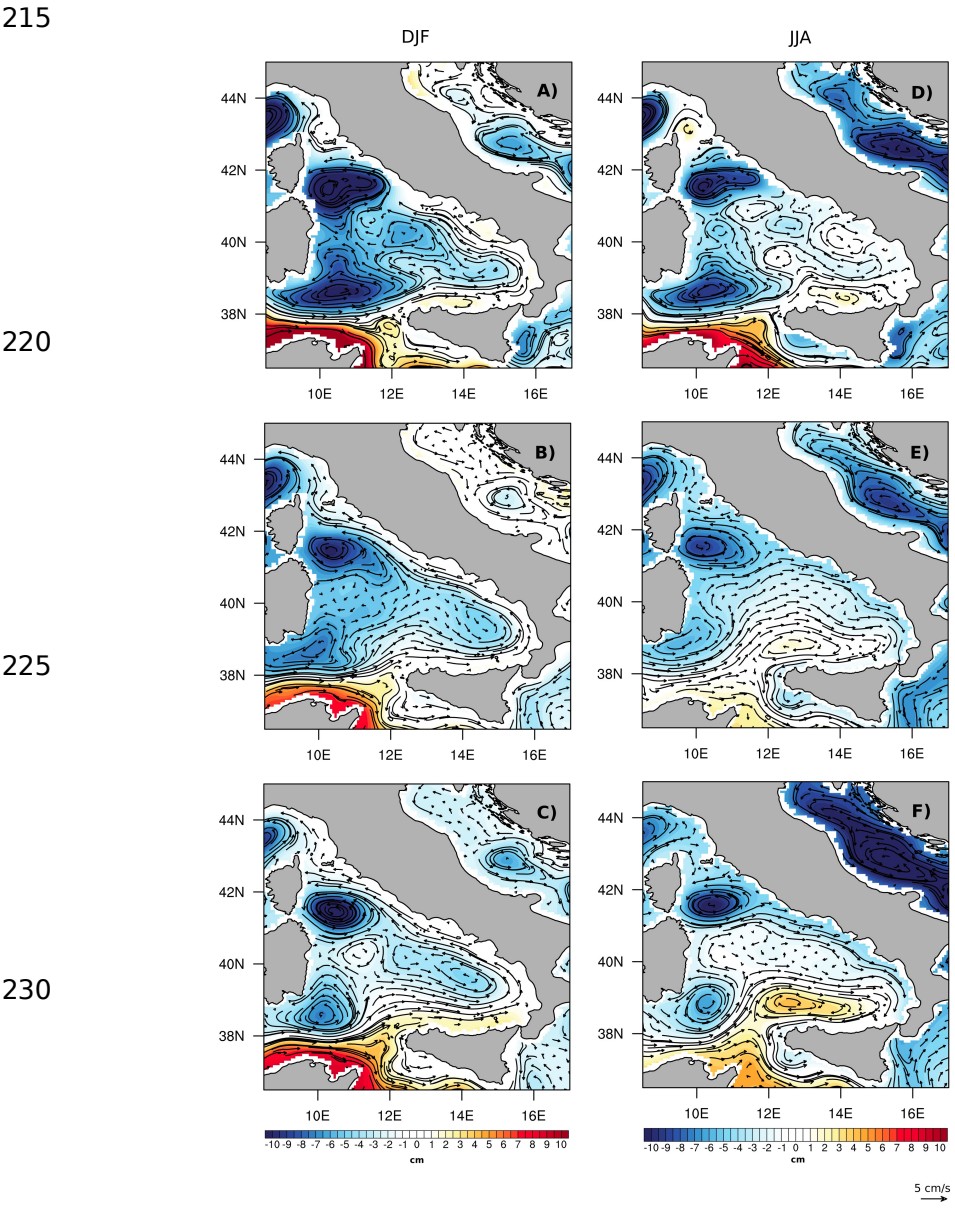

**Figure 3: Winter (left column) and summer (right column) averages of geostrophic circulation (arrows, cm/s) and sea-surface height (SSH, colors, cm) computed (A, D) from AVISO, (B, E) present-day (1976–2005) and (C, F) future (2070-2099) with ROM.**



### 3.3. Impact of climate change on the boundary conditions at the Tyrrhenian straits

Here we focus on the impact of climate change in the current velocity across the Sardinia Strait and the Corsica Channel. Figs. 4 and 5 show vertical cross-sections of current ocean velocity for the Sardinia Strait and the Corsica Channel, respectively, for present-day and future climate. Starting the analysis from the Sardinia Strait, it is important to recall that, at present, in winter, MAW enters the Tyrrhenian to the south of this strait, completes a cyclonic gyre around the basin and partially exits the Tyrrhenian via its northern portion (Figs. 3A-3C). This explains the side-by-side inflow and outflow at

shallow levels (Fig. 4A; see e.g., Vetrano et al., 2004; de la Vara et al., 2019). The MAW (sigma=27.00-28.90; Serravall and Cristofalo 1998) jet stream flows into the Tyrrhenian from the surface to about 150-250 m depth with velocities between 0.10 and 0.15 m/s, while flows out with velocities close to 0.05 m/s. Below MAW, LIW (sigma=28.85-29.10; Lascaratos et al., 1993; Hayes et al., 2019) flows out of the Tyrrhenian (negative values) with the largest speeds of around 0.05 m/s at 250-350 m depth. In the future climate there is a decrease in the water density, more remarkable at the surface, due to the impact

of global warming in water masses characteristics. The surface velocities decrease, with a weakening of both the MAW inflow and outflow, but the LIW outflow to the north of the Strait intensifies with speeds over 0.05 m/s (Fig. 4B). In other words, surface currents associated with MAW weaken, whilst intermediate-depth currents associated with LIW transport intensify. The weakening of the surface flow is associated with a flattening of the isopycnals and remarkable lightening of surface waters, which leads to an increased stratification, in line with what happens in the Gulf of Lions region (Parras-

Berrocal et al., in preparation). In summer, as observed in Fig. 3, the surface circulation pattern is substantially different from that in winter, with a weaker and shallower MAW stream entering the Tyrrhenian through the center of the strait and outflow currents elsewhere in the studied depth range (Fig. 4C). Also, the water column in the first 200 meters is more stratified and lighter than in winter due to summer insolation. In future climate, current velocities experience a generalized decrease, but there is a notable intensification of velocities in the depth range and location of the present LIW, with a

maximum from about 150 to 400 m depth by the Sardinia shelf slope (Fig. 4D). It should be noted that, regardless of the season, in future climate water density decreases throughout the first 400 meters (Figs. 3A-3D). This is related to the generalized temperature increase in the Mediterranean and the freshening of the MAW (Parras-Berrocal et al., 2020).

    Present-time velocities in the Corsica Channel are positive (outflowing) all over the water column and feature a maximum over 0.06 m/s at the surface in winter (Fig. 5A). This implies that, as seen in Figs. 3A-3C, part of the MAWs that

flow around the Tyrrhenian do not complete the cyclonic gyre around the basin and exit via the Corsica Channel. The core of this outflow is located at the west of the channel within the upper 30 m and velocities decrease gradually with depth. In the future, the same pattern remains but velocities become weaker and the maximum, which is close to 0.04 m/s (Fig. 5B), is not located at the surface but further down close to a depth of 200 m. This is consistent with the results found in the Sardinia Strait i.e., the MAW stream that enters the Tyrrhenian weakens in the future. This is also in line with the weakening of the

cyclonic stream around the Tyrrhenian basin and the enhancement of dynamic structures with a shift towards the summer geostrophic circulation pattern. In summer, at present, as can be seen in Fig. 5C and in line with observations, northward current velocities attain a very small magnitude with values not larger than 0.04 m/s, with the stronger values located at 70-100 m depth. In the future, positive (northward) velocities undergo a strong reduction, remaining at the depth of the channel,





but negative values arise at shallow levels, with values of a magnitude greater than 0.02 m/s (Fig. 5D). Again, density

experiences a future decrease all over the water column.

From these results we can conclude that, in future climate, in winter, MAW enters the Tyrrhenian at a slower pace than at present. This is consistent with the slowdown of the along-slope, cyclonic stream that borders the Tyrrhenian in this season, as seen in Figs. 3A-3C and Figs. 4A and 4B. This is also the case for the MAW entering Liguro-Provençal basin via the Corsica Channel, as illustrated in Figs. 5A and 5B. This northward branch of the stream that flows close to the Corsica

coast weakens significantly in the future.

We now examine the changes in the water properties i.e., temperature and salinity across the Sardinia Strait and Corsica Channel (Figs. 6-9). Presently, winter temperatures in the upper 200 m of the Sardinia Strait vary between 15.5ºC to the south to 14ºC to the north (Fig. 6A). This meridional temperature gradient relates to the fact that MAW enters the Tyrrhenian to the south and exits the basin to the north after completing a cyclonic gyre. Down from that depth, temperature

is quite homogeneous and exhibits values close to about 14ºC. In summer, the water column is, as expected, stratified due to the development of the seasonal thermocline, but the meridional temperature gradient is less marked due to the decrease in the MAW inflow into the Tyrrhenian. Values are close to 20ºC at the surface and about 14ºC at a depth close to 100 m (Fig. 6C). From that depth downwards, temperature is close to 14ºC or less. In the future, the winter and summer vertical structure of the water column is qualitatively similar to that at present, but with warmer temperatures (Figs. 6B and 6D). In winter,

surface temperature ranges between 17.5 and 17ºC at the surface and from 150 to 200 m depth water temperature is close to 16ºC. In summer, surface temperature is even warmer than 22ºC. From that depth downwards, temperature is about 16ºC.

At present, in the Corsica Channel, in winter, temperature is quite homogeneous and takes values close to 14ºC with values slightly higher at the surface (Fig. 7A). In the future, winter temperature is homogeneous as well, but shows a nearly 2ºC increase all over the water column (Fig. 7B). In summer, the water column is thermally stratified with values close to

20ºC at the surface and close to 14ºC below the thermocline down to the seafloor (Fig. 7C). In the future summer, the water column remains stratified but warmer, with values close to 24ºC at the surface and values close to 16ºC from about 100 m depth (Fig. 7D).

Focusing now on salinity we observe that, at present, in the Sardinia Strait, salinity is lower at the surface, with a marked north-south gradient and is more homogeneous from about 250 m (Fig. 8A). Specifically, in winter, surface salinity

is close to 37.5 psu to the south and closer to 38 psu to the north of the strait. These results are in line with those from Beranger et al. (2004) and de la Vara et al. (2019). This salinity gradient reflects a side-by-side channel flow which is due to the entrance of fresher MAW from the Algerian Current to the south of the Tyrrhenian which completes a cyclonic gyre around the basin and exits the Tyrrhenian to the north of the strait with a higher salinity. Further deep, from about 250 m, salinity is close to 38.4 psu, which is typical from LIW (e.g., Astraldi et al., 2002). In the future, the winter salinity pattern is

qualitatively similar to that at present, but values are different (Fig. 8B). On the one hand, MAW that enters and exits the Tyrrhenian, respectively, are slightly fresher than today, which may be related to the North Atlantic influence as previously





has reported in Parras-Berrocal et al. (2020) and Soto-Navaro et al. (2020). On the other hand, LIW is saltier than today, with a LIW core with a salinity close to 38.8 psu.

In summer, at present, the entrance of MAW to the Tyrrhenian via the Sardinia Strait is more limited and the

cyclonic stream around the basin is limited to the southern Tyrrhenian Sea, so that the north-south salinity gradient across the Sardinia Strait is more modest than in winter and thus isohalines are less tilted i.e., more horizontal (Fig. 8C). From about 150 m depth, salinity varies less and takes values close to 38.4 psu. In the future, MAW also becomes fresher than at present. LIW becomes, as in winter, saltier than today, with values close to 38.8 psu for the LIW core (Fig. 8D).

In the Corsica Channel, at present, in the winter situation, salinity is lowest at the surface and increases with depth

(Fig. 9A). Its value is close to 38 psu at the surface (see e.g., Béranger et al., 2005; de la Vara et al., 2019) and close to 38.4 psu at the bottom of the channel, which are values close to those from Béranger et al. (2005). In the future, salinity stays similar or lower at the surface, but waters become saltier from about 100 m to the seafloor, reaching up to 38.8 psu (Fig. 9B). In summer, the situation is qualitatively like that reported for the winter season (Figs. 9C and 9D). These changes in salinity make a strong contribution to the reinforcement of the stratification of the waters that enter the Liguro-Provençal basin, as

will be tackled in the Discussion.

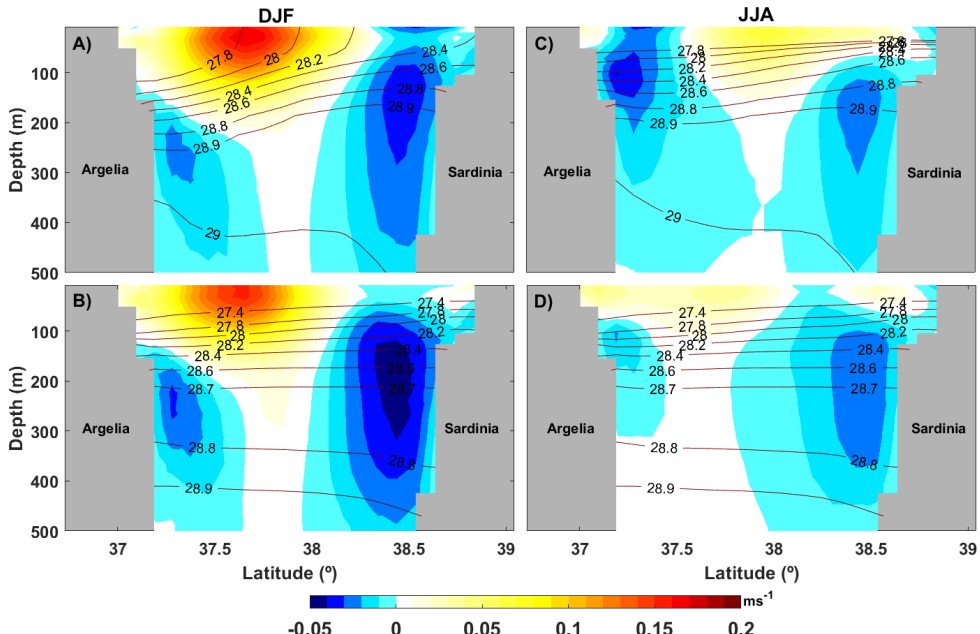

**Figure 4: Winter (left column) and summer (right column) vertical cross-sections of current velocity across the Sardinia Strait (8.04°E and 36.73°N to 8.79°E to 39.04°N; see Fig. 1), in m/s, for present (A and C) and future (B and D) climate as computed from ROM. Positive values indicate eastward flow and negative values the opposite. Isopycnals are also depicted. Note that only the upper 500 m of the water column are shown for a more detailed**

**visualization of surface velocities.**





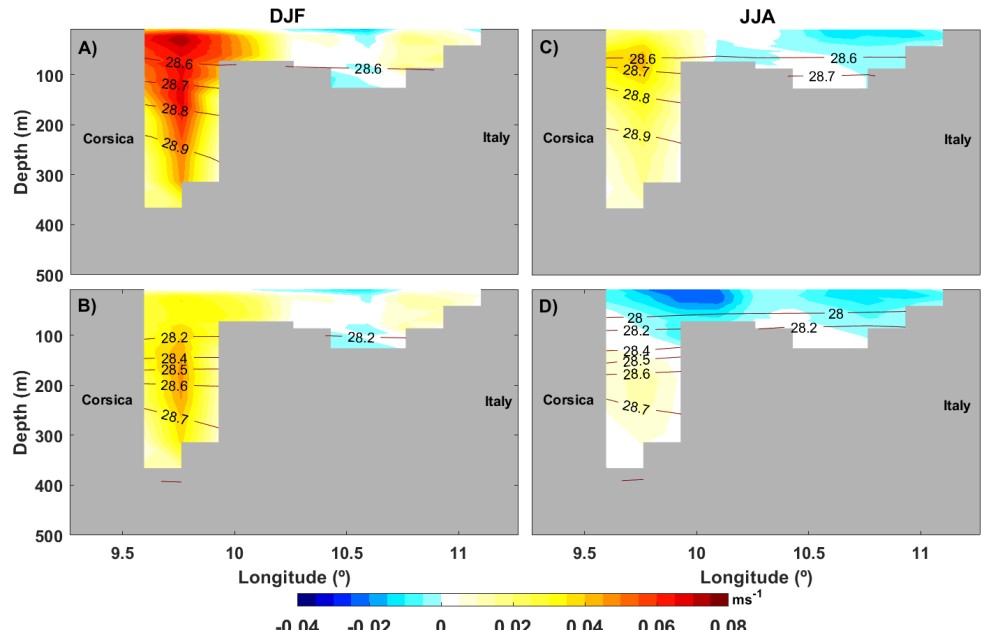

**Figure 5: Winter (left column) and summer (right column) vertical cross-sections of current velocity across the Corsica Channel (9.26°E and 42.53°N to 11.26°E and 42.80°N; see Fig. 1), in m/s, for present (A and C) and future (B and D) climate as computed from ROM. Positive values indicate northward flow and negative values the opposite. Isopycnals are also depicted.**





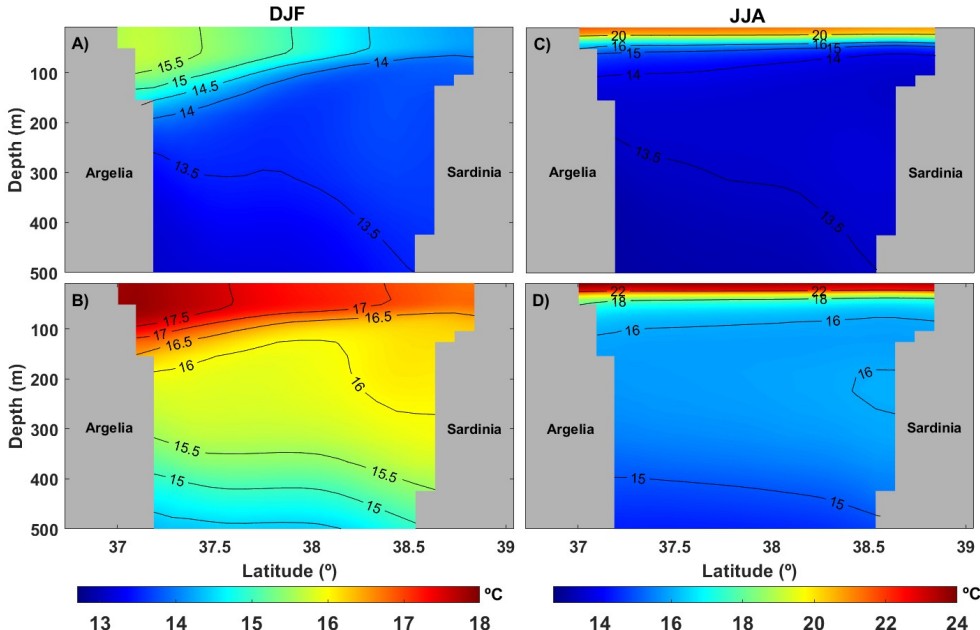

**Figure 6: Winter (left column) and summer (right column) vertical cross-sections of temperature, in ºC, through the Sardinia Strait for present (A and C) and future (B and D) climate as computed from ROM. Isotherms are also depicted.**




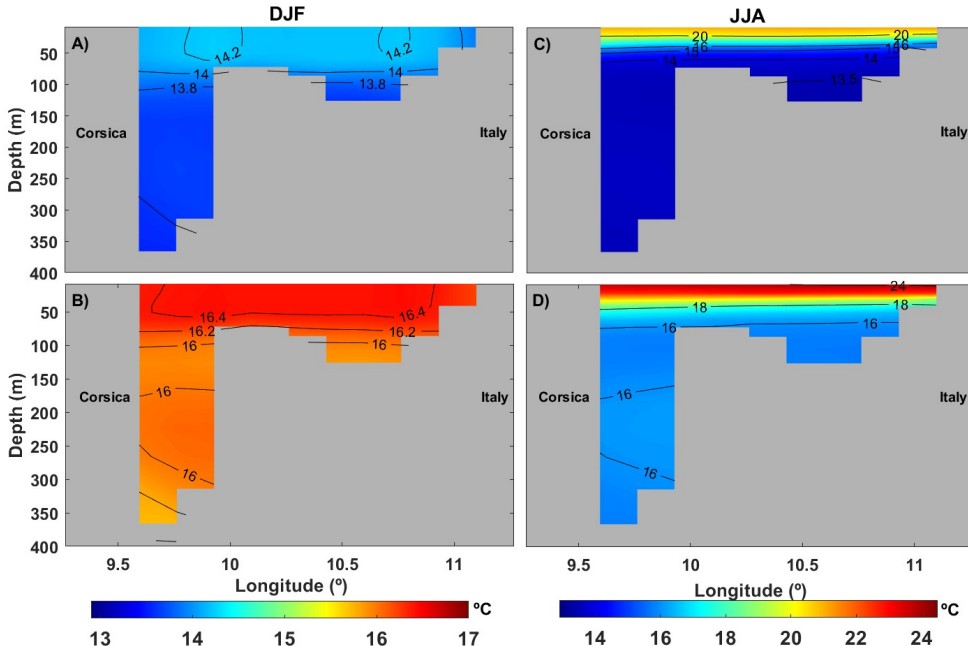

**Figure 7: This is equivalent to Fig. 6, but constructed for the Corsica Channel.**



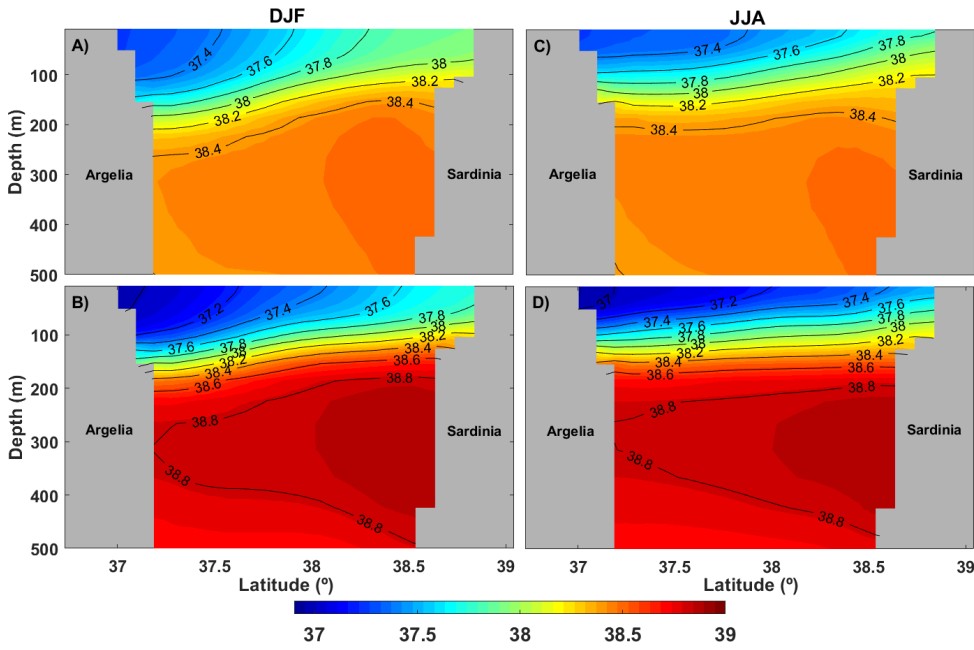

**Figure 8: Winter (left column) and summer (right column) vertical cross-sections of salinity, in psu, through the Sardinia Strait for present (A and C) and future (B and D) climate as computed from ROM. Isohalines are shown.**



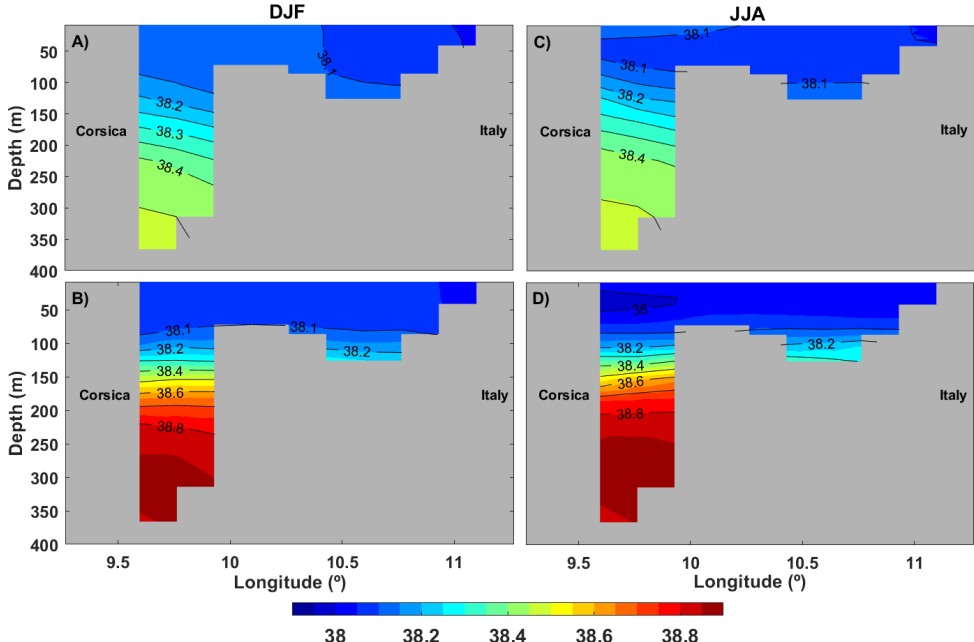

**Figure 9: This is equivalent to Fig. 8, but constructed for the Corsica Channel.**

## 4. Discussion

At this point it is interesting to understand the causes and consequences of the projected changes in the Tyrrhenian
Sea geostrophic circulation and water transport across its main straits.

### 4.1. Drivers for the future changes in the geostrophic circulation of the Tyrrhenian

Here we investigate the role of winds in the changes of the geostrophic circulation projected for the end of the
century, as seen in the previous section. The surface wind pattern over the Tyrrhenian Sea presents a clear seasonality. We
observe that in present time winter the there is a predominance of westerly winds (Fig. 10A). The strongest wind speeds
occur to the east side of Sardinia and Bonifacio straits, due to wind funneling (see e.g., Gérigny et al., 2015). These areas of
enhanced wind speed coincide with those where the main dynamical structures are found (see Figs. 3A-3C and e.g., de la
Vara et al., 2019). The lowest wind speeds occur near coastal areas, with values even below 1 m/s. Whilst wind direction
does not display important differences in the future, its module shows a generalized decrease in the future, but specially in
the Sardinia and Bonifacio Straits, related mainly to a speed decrease, with a reduction of the largest values from 6 m/s to





values between 4 and 5 m/s (Fig. 10B). The wind slackening found near the entrance of the Sardinia Strait may be an important trigger for the slowdown of the cyclonic stream around the basin which is fed by MAW.

In summer, winds are far weaker than in winter, with an overall decrease of the magnitude of the westerlies (Fig. 10C). Once more, the greatest wind speeds arise to the east of the Bonifacio Strait, with values close to 2 m/s. In this season, high wind speeds also occur to the southeast of the Tyrrhenian, where two important dynamical structures are found (see Figs. 3D and 3E). As found for the winter season, the future pattern of surface wind is qualitatively similar to the present one, but with remarkably lower wind speeds (Fig. 10D).

We now study the relationship between the wind representation and the ocean circulation. To that end, we compute the mean wind work in the Tyrrhenian e.g., Renault et al., 2016). This is calculated using the following expression:

$$Windwork = \frac{1}{\rho_o}\left(\overline{\tau_x}\cdot\overline{u_o}+\overline{\tau_y}\cdot\overline{v_o}\right),$$

where $\rho_o$ is the mean value of the sea water density assumed 1020 kg/m$^3$, $u_o$ and $v_o$ the zonal and meridional components of the ocean velocity of the five more superficial ocean model levels (down to 51 m) in m/s and $\tau_x$ and $\tau_y$ represent the zonal and meridional components of the wind stress, respectively, in N/m$^2$. The overbars indicate climatological averages estimated for the present and future time periods. The wind work represents the exchange of kinetic energy between the atmosphere and the ocean. Positive values indicate that both winds and ocean currents move in the same direction i.e., wind provides kinetic energy to the ocean. Negative values, however, indicate that they move in different directions and thus that winds reduce the ocean's kinetic energy. Positive and negative side-by-side values either correspond to water recirculation or the presence of an eddy. Overall, as shown in Figs. 11A and 11C, the winter and summer patterns found for present-day mean wind work in the Tyrrhenian are in line with those found in de la Vara et al. (2019). In winter, the combination of westward winds and a cyclonic stream around the Tyrrhenian leads to positive values from the Sardinia Strait to the easternmost part of the basin (Fig. 11A). Then, negative values arise as the water continues to flow to the northwest of the Tyrrhenian Sea. Side-by-side positive and negative values show up to the east of the Bonifacio Strait, where the Bonifacio Gyre is found. Negative values are also present in the recirculation area near the Sardinia Strait. In the future, qualitatively, the same basin-scale pattern remains, but positive and negative values of the wind work become remarkably higher in the Bonifacio Gyre with up to 8 cm$^3$/s$^3$ in that region (Fig. 11B). The future enhancement of the wind work, especially in the areas where the dynamical structures appear is thus not caused by wind speed intensification. The enhancement of the wind work may be related to the weakening of the cyclonic stream around the Tyrrhenian, which makes stronger the transference of kinetic energy from the wind to the Bonifacio Gyre and could be responsible for its intensification and increased extension. The strength of the Bonifacio Gyre is determined by the energy input by the wind stress and its interaction with the cyclonic jet, which employs part of the wind energy for the shedding of smaller-scale eddies, constraining the size of the gyre. In the future, the weakening of the stream makes it more stable, reducing the shedding, which is reflected by the increase in size of the Bonifacio Gyre and its strengthening.





As shown in Fig. 11C, at present, the wind work in summer is mainly positive except for the southeast of the Tyrrhenian, where positive and negative side-by-side values appear due to the development of a characteristic summer
dynamical structure and in the Bonifacio Gyre (Figs. 3D-3F; e.g., de la Vara et al., 2019; Napolitano et al., 2016). It takes smaller values than in winter due to the wind slackening. In the future, the wind work does not present important changes, but increases especially in the Bonifacio Gyre area (Fig. 11D). The reasoning behind this is that, again, a decrease in the water transport of MAW across the Sardinia Strait increases the effectiveness of the transference of kinetic energy from the wind to these dynamical structures.

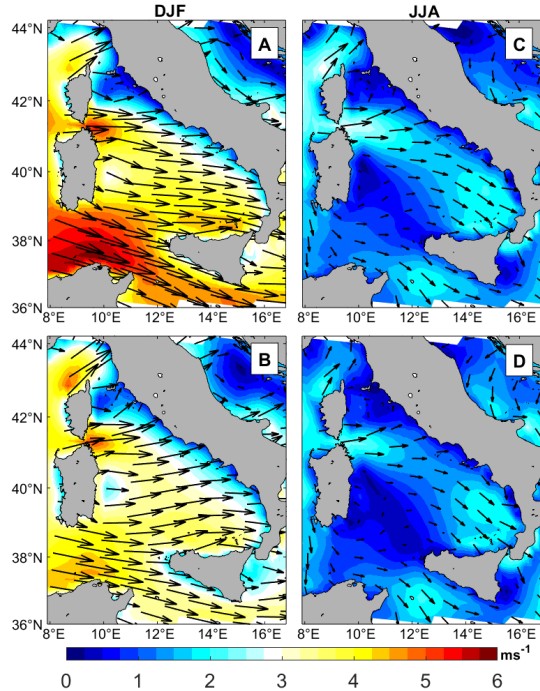

**Figure 10: Winter (left column) and summer (right column) averages of the wind speed module (colors, m/s) and direction (arrows) for present (A and C) and future (B and D) climate as computed from ROM.**


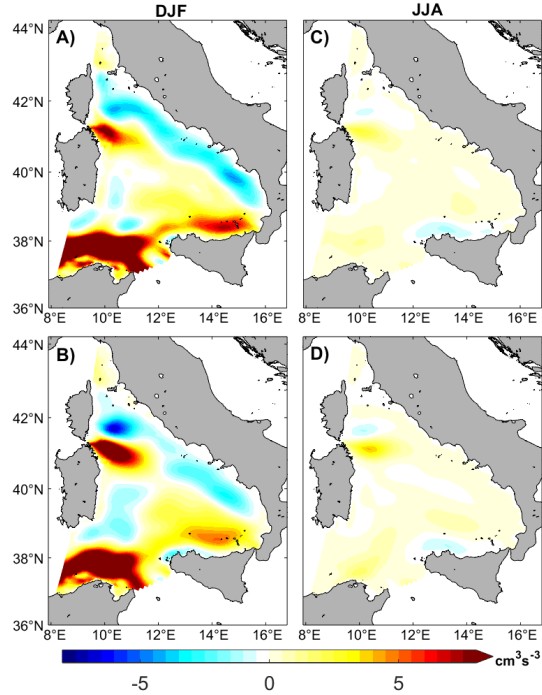

**Figure 11: Winter (left column) and summer (right column) averages of the wind work (cm$^3$/s$^3$) taking into account ocean currents averaged from the surface to a depth of 51 m for present (A and C) and future (B and D) climate as**
**computed from ROM.**

## 4.2. Implications for the water exchange with other basins

We now study how the changes in the water properties across the Tyrrhenian straits in the future can have a possible impact on adjacent areas. In particular, we focus on the Liguro-Provençal basin, the circulation and hydrology of
which is strongly affected by waters that exit the Tyrrhenian Sea via the Corsica Channel and this can have a concomitant effect on the deep convection over the Gulf of Lions (Astraldi et al., 1999).

From the above analysis we can thus conclude that in both the Sardinia Strait and the Corsica Channel the water column presents, qualitatively, the same structure as today but waters experience a generalized warming which is in line with Parras et al. (in preparation). Also in agreement with that work, surface waters tend to be fresher than today, whilst
intermediate-depth waters will be saltier, due a freshening of the Atlantic surface waters that enter the Mediterranean, as well as by an enhanced evaporation that causes intermediate waters to be saltier than today. The prevailing effect of the warning leads a generalized density decrease all over the sections studied (Figs. 4 and 5). This reduction is more pronounced at the surface due to the simultaneous warming and freshening, and at intermediate-depths, the density reduction is more modest because both temperature and salinity show an increase. A clear implication of this is that the water column becomes more
stratified in the future. The waters flowing from the Tyrrhenian towards the Liguro-Provençal basin play a first-order



importance role in shaping the stratification of the area and thus in the preconditioning for deep water convection in the Gulf of Lions (e.g., Schroeder et al., 2010). Our results suggest that the combination of the decrease of the volume and the strengthening of the stratification of the waters that flow through the Corsica Channel towards the Liguro-Provençal basin could be one of the contributing factors to the projected interruption of the deep water formation in this area in the future 425 (Margirier et al., 2020; Parras-Berrocal et al. (in preparation)).

## 5. Conclusions

In this work we examine the possible changes in the Tyrrhenian future surface circulation under the RCP8.5 emission scenario with the atmosphere-ocean regionally coupled model ROM. Special attention is given to the mechanisms that drive the changes by the end of the 21$^{st}$ century and the possible implications in the Liguro-Provençal basin. The 430 Tyrrhenian Sea is a challenging region to study the impact of global warming on the oceanic circulation: it presents complex seasonal-dependent circulation patterns and rich mesoscale activity which is controlled by the interaction of local climate and the large-scale Mediterranean circulation through the water transport at its main straits. We show that ROM is capable of successfully reproducing the main features of this circulation. The main conclusions of this study regarding circulation changes in future can be summarized as follows:

- In winter, the cyclonic stream around the Tyrrhenian becomes weaker compared to that at present due to a reduced water transport across the Sardinia Strait, while the Bonifacio Gyre and the recirculation area near the Sardinia Strait intensify. This means that the winter pattern becomes less developed than at present, which is consistent with a warming climate.

- In summer, the dynamical structures become more intense than at present and, as today, the cyclonic stream around 440 the Tyrrhenian basin breaks down. Therefore, the summer pattern becomes more intensified than at present.

- The intensification of the dynamical structures in winter is related to the weakening of the cyclonic stream around the Tyrrhenian Sea which is, in turn, caused by a reduction in the inflow of MAW into the basin. This weakening is associated with the increase of the magnitude of the wind work over the main dynamical structures, which are intensified and enlarged.

- In the future, the northward water transport across the Corsica Channel towards the Liguro-Provençal basin becomes smaller than today. Also, water that flows through this channel presents a stronger stratification than at present due to a generalized warming with a saltening of intermediate waters. These changes potentially contribute to the interruption of deep water formation in the Gulf of Lions by the end of the century.




**Data availability**

The ROM data are available at https://swiftbrowser.dkrz.de/public/dkrz_64ea1a99-f1de-45dab8d1-a3175f15ee46/
ROM_MED_dataset/ (Sein et al., 2015).

**Competing interests**

The authors declare that they have no conflict of interest.

**Acknowledgements**

A. de la Vara and W. Cabos have been funded by the Spanish Ministry of Science, Innovation and Universities, the
Spanish State Research Agency and the European Regional Development Fund, through grant CGL2017-89583-R. W.
Cabos has also been supported by Salvador Madariaga grant (Spanish Ministry of Science Innovation and Universities). I. M.
Parras Berrocal was supported through the Spanish National Research Plan through project TRUCO(RTI2018-100865-B-
C22). D. Sein worked in the framework of the state assignment of the Ministry of Science and Higher Education of Russia
(0128-2021-0014).

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
