# Peer review of "Climate change signal in the ocean circulation of the Tyrrhenian Sea"

_Earth System Dynamics, 2021_

## Author Comment (AC1)

*Response to reviewer 1*

**General comment on "Climate change signal in the ocean circulation of the Tyrrhenian Sea" by Alba de la Vara et al.**

Thank you for your positive evaluation on our manuscript. We believe that the raised comments have helped us to attain a more complete and robust version of the manuscript.

**The manuscript addresses future changes that could take place in the Tyrrhenian Sea (TS) circulation by the end of the current century under the "business–as-usual" high-emission RCP8.5 scenario. Authors use the regionally-coupled climate model ROM (from REMO-OASIS-MPIOM) to conclude that surface mesoscale patterns in the basin are slightly modified, which authors asc,ribe to changes in water transport across Sardinia Strait and mechanical energy transfer from wind field to the ocean in the vicinity of Bonifacio Strait. More interesting is the conclusion that the flow from the TS into the Liguro-Provençal basin will weaken and the advected waters will be more stratified, speculating about the possibly hampering of winter deep water formation in the Gulf of Lions by the end of the century. While the objective of the manuscript is of interest, a question that arises is why authors made up their mind to select the TS instead of the critical areas of intermediate and deep water formation, focusing on these important processes. Such a study would provide a deeper insight on the future evolution of the Mediterranean Sea circulation. I would like to have read a sentence explaining the reasons and the pros of their selection against the other alternatives.**

Thank you for the comment. The main reasons to choose the Tyrrhenian Sea are the following. First, it features distinct winter and summer surface circulation patterns with an enriched in dynamical mesoscale-size structures in summer. Thus, from a modeling perspective it is a challenging area and the study of the future evolution of these patterns is of interest. Second, because the Tyrrhenian Sea is connected to the Liguro-Provençal basin and the waters that reach this area from the Tyrrhenian play an important role in the preconditioning for deep water formation in the Gulf of Lions. Therefore, changes in the water properties within the Tyrrhenian Sea inherent to climate change may have an impact on deep winter convection in the Gulf of Lions, which is important for the hydrographic properties of the Mediterranean Sea water, its thermohaline circulation and thus sea-bottom ventilation. Third because, to our knowledge, there are no studies devoted to the study of changes in the Tyrrhenian surface circulation in future climate. This insight will be added to the Introduction, to make it clear our choice to focus on the Tyrrhenian Sea. This being said, we agree that critical areas of intermediate and deep water formation have to be analyzed. However, given the complexity of these water formation processes, they merit a separate in-depth study.

**In its present form, there are some points that should be considered and revised or completed. My objections refer to the model set-up, mainly the (boundary?) condition at Gibraltar, and to the validation. The review only addresses these two aspects and it does not include specific comments, technical corrections or typing errors.**

Thank you for your consideration. The ROM coupled system does not use any boundary conditions at Gibraltar. Actually, the oceanic component of our coupled model is global and the boundaries of the atmospheric component are located far from Gibraltar. As previously exposed in Parras-Berrocal et al. 2020: "ROM introduces the novel approach of implementing a global ocean model with high horizontal resolution at regional scales. This allows us to obtain information of the global ocean maintaining the high spatial resolution in the coupling area. The model simulates explicitly the exchange of water through the Strait of Gibraltar and the Dardanelles, taking into account the signals from the neighboring basins (i.e. Atlantic Ocean), which are essential to include the large-scale feedbacks in the climate signal of the Mediterranean."

In response to this comment, this will be more clearly explained in the model setup in the revised version of the manuscript.

**MODEL SET-UP**

**One of the ms main conclusions is the future enhanced stratification in the TS, which is the joint result of the expected SST increase in a warming ocean and the freshening of the surface water. Whereas the SST increase is an undisputed fact, the freshening is not. Obviously, it must be the result of a fresher Atlantic inflow through the Strait of Gibraltar, which has consequences on the whole Mediterranean Sea, not only in the TS.**

Indeed, the freshening of Atlantic inflow has consequences in the Mediterranean Sea, more evident in the Western Mediterranean. As shown in Parras-Berrocal et al. (2020) the SSS in the Western Mediterranean under the RCP8.5 is expected to slightly reduce (from −0.5 to −1.0 psu; see Figure 1R). The authors point out that this freshening is a direct consequence of the North Atlantic Ocean, accounted through the ROM global ocean component. Moreover, as shown in Soto-Navarro et al. (2020), the freshening of the surface waters in the Western Mediterranean is a robust feature in the coupled regional simulations analyzed in that paper. In response to this comment, the robustness of the results regarding the future signal of the SSS in the Mediterranean Sea will be mentioned in the revised manuscript in Section 3.3.

[Figure]

Figure 1R. Mean SSS (left, in psu), averaged over the 1976–2005 period (ROM). Differences between mean SSS (right, in psu) in RCP8.5 projection (2070–2099) and present climate (1976–2005). Taken from Parras-Berrocal et al. (2020).

This surface freshening together with the expected warming of the whole water column will strengthen the stratification in the Tyrrhenian Sea in the future. As a result, higher values of stratification index (SI) express higher stratification of the water column. In our simulations (Figure 2R), there is a general increase of the vertical stratification in the Tyrrhenian Sea at the end of the 21st century, especially in summer.

[Figure]

Figure 2R. Stratification Index map (in $m^2/s^2$) averaged for the periods (i) 1976-2005 and (ii) 2070-2099 computed from the surface to 51 m depth.

**And obviously again, all the results of the modelling (not in the TS uniquely) rely critically on the validity of this boundary condition. Except for a vague sentence on lines 104-105 ("the water exchange at Gibraltar and Dardanelles in ROM is not parameterized and Atlantic water properties are not relaxed towards climatological values in the areas adjacent to the straits"), nothing else is said about this critical point, as if the future freshening of the Atlantic inflow were a proven result or another indisputable fact. Does this result come from MPIOM global model? Authors must be much more explicit about this point, discuss its validity and assess how sensitive their conclusions are against small changes of Atlantic inflow salinity.**

This comment has already been addressed in the previous response. As pointed out by the Reviewer, the freshening of Atlantic inflow comes from the global oceanic component of ROM (MPIOM). Soto-Navarro et al. (2020) also found a freshening of the Western Mediterranean under RCP8.5 scenario in several Regional Climate Models and their explanation for the SSS reduction in the Western Mediterranean is related to the Atlantic conditions. Furthermore, they also pointed that "different models advect the fresher waters from the Arctic following different paths, and in some cases they may arrive close to the Iberian Peninsula and thus affect the waters entering into the Mediterranean, as seen by Gomis et al. (2016)". In response to this comment, we will expand this issue further in the revised manuscript.

**Similar considerations can be extended to the strength and size of the exchanged flows through the Strait of Gibraltar, which should be notably increased if the density contrast between inflow (warmer and fresher) and outflow increases. Could authors show the ROM forecast for these variables in the "Model setup" section and extend the section by addressing the conditions that hold at the western boundary of the Mediterranean. It is necessary in order to support adequately the conclusions of the paper and to discuss how robust they are against changes of those conditions.**

Thank you for the comment. As we explained above, ROM does not use boundary conditions for the Mediterranean Sea, and the exchange flows at the Strait of Gibraltar are explicitly simulated, obviously

accounting for changes in surface and internal pressure gradients originating from changes in SSH and water masses characteristics, with a resulting net water flow that must compensate the net evaporation (Evaporation - Precipitation -River Runoff) over the whole Mediterranean. Indeed, as pointed out by the Reviewer, the density contrast between inflow and outflow increases by 0.2 kgm$^{-3}$ (nearly 10% of the density contrast) from the 2006-2040 (1026.8 kgm$^{-3}$ for inflow and 1028.7 kgm$^{-3}$ for outflow) to the 2070-2099 (1026.3 kgm$^{-3}$ for inflow and 1028.4 kgm$^{-3}$ for outflow) period. As a result, there is a slight reduction in both water inflow and outflow, but with a noticeable increase in the net water flow (from 0.037 Sv in 2006-2040 to 0.056 Sv in 2070-2099) in order to compensate for the increased net evaporation (E-P+R) over the Mediterranean basin in the 2070-2099 period. This point will be clarified in the revised manuscript, where it will be stressed that no boundary conditions are applied at the Strait of Gibraltar.

**VALIDATION is addressed in a very light way in Section 3 (RESULTS), where it appears in different places along with the interpretation of the model outputs. AVISO geostrophic currents is the reference for validation, which is carried out by comparing the thirty-year period 1976-2005 averaged circulation from ROM with the thirteen-year (1993-2005) period of AVISO.**

Many aspects of the simulations analyzed here have been validated in Parras-Berrocal et al (2020). However, for the sake of completeness, the validation of present time results will be expanded in the revised version of the manuscript, both the parts regarding the Mediterranean and the Tyrrhenian geostrophic circulation.

The reason why the time periods considered for the validation are not exactly the same is because AVISO data are only available since 1993. In order to address the reviewers' concerns, we have plotted some of the validation figures for the suggested time period. The geostrophic circulation of the Mediterranean Sea and the Tyrrhenian Sea computed from ROM and AVISO considering the 1993-2005 time period is represented in Figures 3R and 4R. As it can be observed, differences between the results obtained with ROM considering the 1993-2005 or the 1976-2005 time period are small and do not influence qualitatively our findings. Thus, for simplicity and consistency with figures shown in the Results, which have to be necessarily created with data from the 30 years 1976-2005 period, as recommended in the World Meteorological Organization Guide nº100, we will keep the latter period for the validation of the present-day Mediterranean and Tyrrhenian Sea, respectively. In response to this comment, we will mention the reasons why the AVISO and ROM data time periods are not exactly the same, and the consequences of our choice in terms of the results will be stated in the figure captions of Figures 2 and 3 of the manuscript.

[Figure]

Figure 3R. Winter (left column) and summer (right column) averages of present-day Mediterranean geostrophic circulation (vectors, cm/s) and sea-surface height (SSH, colors, cm) from ROM (first row), AVISO (second row) and a CMEMS* reanalysis (third row). Results are computed with data from the 1993-2005 time period. Only one out of four vectors is plotted.

*The CMEMS reanalysis used for the revision is MEDSEA_MULTIYEAR_PHY_006_004, with a horizontal resolution of 4-5km (see https://resources.marine.copernicus.eu/?option=com_csw&view=details&product_id=MEDSEA_MULTIYEAR_PHY_006_004).

[Figure]

Figure 4R. Winter (upper row) and summer (lower row) averages of present-day Tyrrhenian geostrophic circulation (vectors, cm/s) and sea-surface height (SSH, colors, cm) from ROM (first column), AVISO (second column) and and a CMEMS reanalysis (third column). Results are computed with data from the 1993-2005 time period. Only one out of four vectors is plotted.

**From Figure 2, it is difficult to conclude that both patterns agree satisfactorily, partially because of the poor resolution of the figure. Rather, the agreement seems to be only moderate if not arguable. But, even if it was satisfactory, the question that arises is why different time periods are used to average both data when ROM can use exactly the same period as AVISO. Why this mismatch of periods? Considering that the longer the averaging period the smoother the resulting pattern, the, for instance, disagreement between the mesoscale-rich circulation summer pattern in the TS from AVISO (Figure 3D) and ROM (Figure 3E) could be partially explained. In any case, the right way of comparing results is to make use of identical periods if possible.**

As mentioned in above responses, the text regarding the validation of the Mediterranean and Tyrrhenian geostrophic circulation will be expanded. Also, the point regarding the different time periods considered for the evaluation and the changes that will be introduced in the manuscript were tackled in the previous response.

**Reasons provided in lines 136-140 to focus on the surface geostrophic circulation for validation purposes are strange. Actually, using surface geostrophic circulation and/or sea surface height (SSH) from AVISO is out of necessity: they are the only available variables.**

Thank you for your comment. We are not sure we understand your point but, for the sake of clarity, we prefer to keep the text as it was in the originally-submitted version of the manuscript, but stating that we use the Mediterranean version of the AVISO dataset.

**The mention to the in-preparation paper by Parras-Berrocal et al. in order to justify the realistic representation of the main features of the surface circulation (lines 156-157) should be removed. If ROM is successful in doing this, the results should be either shown in this paper or referred to an already published paper, not to an in-preparation work.**

Thank you for your recommendation. We will replace Parras-Berrocal et al. (in preparation) by Parras-Berrocal et al. (2020). There, the authors validate the ability of ROM to reproduce the Mediterranean sea-surface height for the 1980-2012 using AVISO. It is shown that ROM is able to successfully reproduce the main circulation structures in the Western Mediterranean (see Figures 11 and 12 of Parras-Berrocal et al. 2020).

**Reference list**

Gomis, D., Alvarez-Fanjul, E., Jordà, G., Marcos, M., Martínez-Asensio, A., Llasses, J. and Sotillo, M. G.: Regional marine climate scenarios in the NE Atlantic sector close to the Spanish shores. Sci Mar., 80:215–234, doi: 10.3989/scimar.04328.07A, 2016.

Parras-Berrocal, I., Vazquez, R., Cabos, W., Sein, D., Mañanes, R, Perez-Sanz, J., and Izquierdo, A.: The climate change signal in the Mediterranean Sea in a regionally coupled ocean-atmosphere model, Ocean Sci., doi:10.5194/os-2019-42, 2020.

Soto-Navarro, J., Jordá, G., Amores, A., Cabos, W., Somot, S., Sevault, F., et al.: Evolution of Mediterranean Sea water properties under climate change scenarios in the Med-CORDEX ensemble, Clim. Dyn., 54(3), 2135-2165, doi:10.1007/s00382-019-05105-4, 2020.

---

## Author Comment (AC2)

**Response to reviewer 2**

**Review of the manuscript "Climate change signal in the ocean circulation of the Tyrrhenian Sea", by Alba de la Vara et al.**

Thank you for your positive evaluation of our manuscript. The posted comments have helped us to achieve a more complete and robust version of the manuscript.

This work is devoted to the analysis of the future evolution of the circulation in the Tyrrhenian Sea (TS) under the pessimistic RCP 8.5 emission scenario. Using the outputs of the regional atmospheric-ocean coupled model ROM, the authors study the projected changes in the seasonal circulation patterns and the exchange through the main straits connecting the TS with the rest of the Western Mediterranean: the Sardinia Strait (SS) and the Corsica Channel (CS). The results show a weakening of the TS main cyclonic circulation and an enhancement of the mesoscale structures of this sub-basin. Authors attribute this changes to a reduced inflow of Modified Atlantic Waters across the SS and to an increase of the kinetic energy transferred by the wind. In addition, the results also show an increase of the water column stratification and a reduction in the water transport to the Liguro-Provençal basin across the CS, which may impact the deep water formation process at the Gulf of Lions.

For my review, I'll take advantage of the journal's format that allows me to read the comments of my fellow reviewer. I agree with him/her in the interest of this work, but I'm also surprised that the authors have chosen the TS as area of study. Given the potential of the model to reproduce complex dynamics processes in the whole Mediterranean, I coincide with reviewer #1's (R1) opinion that extending the region analyzed to, at least, include key processes like the deep water formation would have substantially increased the impact of the paper. Clarifying the reasons of this choice is important in my view.

Thank you for this comment. As exposed in the response to Reviewer 1, the main reasons to choose the Tyrrhenian Sea are the following. First, it features distinct winter and summer surface circulation patterns with an enriched in dynamical mesoscale-size structures in summer. Thus, from a modeling perspective it is a challenging area and the study of the future evolution of these patterns is of interest. Second, because the Tyrrhenian Sea is connected to the Liguro-Provençal basin and the waters that reach this area from the Tyrrhenian play an important role in the preconditioning for deep water formation in the Gulf of Lions. Therefore, changes in the water properties within the Tyrrhenian Sea inherent to climate change may have an impact on deep winter convection in the Gulf of Lions, which is important for the hydrographic properties of the Mediterranean Sea water, its thermohaline circulation and thus sea-bottom ventilation. Third because, to our knowledge, there are no studies devoted to the study of changes in the Tyrrhenian surface circulation in future climate. This insight will be added to the Introduction, to make it clear our choice to focus on the Tyrrhenian Sea. This being said, we agree that given the potential of the model critical areas of intermediate and deep water formation have to be analyzed. However, given the complexity of these water formation processes, they merit a separate in-depth study.

Nonetheless, the results obtained are relevant and well presented, and the conclusions are well-argued. However, I have some concerns about both formal and content aspects of the current MS.

**Formal aspects:**

My main recommendation here is that the differences between the future and present climates would be much easier to understand if they are directly represented in the figures. Instead of only plotting the average for the present and future, if you include a panel with their differences with a positive-negative color bar the reader will be able to rapidly identify the regions where the different magnitudes increase/decrease. I understand that this is tricky for the velocity vectors, but for the scalar magnitudes it would be of great help.

In response to this comment, Figures 4-11 have been updated to additionally include the differences between the future and present climate situation of the corresponding fields. The corresponding figure captions have also been accordingly modified.

**Contents:**

Model set-up: In general, I agree with R1 that the description of the model should be extended, more considering that one of the key results strongly depends on the modification of the MAW properties. It is important to understand how the Atlantic Waters are imported through the Strait of Gibraltar and which are the conditioning factors for their lower salinity. For instance, is the salinity of the GCM used as boundary condition for ROM, MPI-ESM-MR, driving the properties of the inflowing waters?

The oceanic component of ROM, MPIOM, is global and the use of any boundary conditions is not necessary. Therefore, the exchange flows through Gibraltar are explicitly simulated by our regionally coupled model. As the atmospheric domain is large and encompasses most of the North Atlantic, MPIESM influences the oceanic properties through the large-scale forcing on the atmospheric component of ROM. This aspect will be extended in the Model Setup of the revised version of the manuscript.

Validation: Following my previous comment, Parras-Berrocal et al. (2020) show that for a hindcast of ROM forced by ERA-Interim in the present climate the average SSS in the Mediterranean is between 1 and 2 psu higher than MPI-ESM. This result should be commented here or in the previous section to contextualize the results. It means that, for the present climate, the GCM could be underestimating the salinity. This doesn't necessarily mean that the projected freshening of the surface layer is wrong, but it is an important information in order to interpret the results. Comparing with the SSS projected by other regional models of the Mediterranean, particularly if they are forced by different GCMs, would also give more context to the results.

Regarding the differences in SSS between ROM and MPIESM, in Figures 4R and 5R, in response to a later comment, we compare in detail the SSS simulated by ROM with a high-resolution reanalysis from CMEMS and MPIESM in both the Mediterranean and the Tyrrhenian Sea. In the Mediterranean Sea, the global oceanic component of ROM (MPIOM) improves the SSS simulated by MPIESM relative to the high-resolution reanalysis.

The salinity simulated by ROM when forced by MPIESM has been validated individually in Parras-Berrocal et al. (2020) and in a multimodel study by Soto-Navarro et al. (2020). In Soto-Navarro et al. (2020) it is shown that in the present-time evaluation period (1987-2005) the ROM performance is similar to that of the other state-of-the-art regional coupled models participating in the MEDCORDEX project. Although the amplitude of the SSS seasonal cycle in ROM was lower than in the datasets used for evaluation, the amplitude of the seasonal cycle in the upper layer (0-150 m) as well as the standard deviation of SSS and the whole water column salinity in ROM is close to the range of the datasets. In Section 3.3 of the revised manuscript we will add the following text: "We would like to emphasize that a detailed validation of ROM's capability to simulate the present-time salinity and temperature over the Mediterranean Sea has been extensively studied (we refer the reader to Darmaraki et al. (2019); Parras-Berrocal et al. (2020) and Soto-Navarro et al. (2020) for details). In particular, ROM improves a strong negative bias in SSS present in MPI-ESM."

**In the validation of the geostrophic currents using AVISO altimetry data, I also agree with R1 that the results shown in figure 2 are not as conclusive as the authors claim, and that the comparison should be made using the same periods for the average.**

As stated in the response to Reviewer 1, many aspects of the simulations analyzed here have been validated in Parras-Berrocal et al. (2020). However, for the sake of completeness, the validation of present-time results will be expanded in the revised version of the manuscript, both the parts regarding the Mediterranean and the Tyrrhenian geostrophic circulation.

The reason why the time periods considered for the validation are not exactly the same is because AVISO data are only available since 1993. We took 1976-2005 because 30-year time periods are recommended by the World Meteorological Organization for validation and 1976-2005 is the standard for the historical run validation in CMIP5. Nevertheless, we computed the geostrophic circulation of the Mediterranean Sea and the Tyrrhenian Sea for ROM, a CMEMS\* reanalysis and AVISO considering the 1993-2005 time period (Figures 1R and 2R). As it can be observed, differences between the results obtained with ROM considering the 1993-2005 or the 1976-2005 time period are small and do not change qualitatively our findings. Thus, for the sake of simplicity and consistency with figures shown in the Results, which have to be necessarily created with data from the 1976-2005 period, we will keep the latter period for the validation of the present-day Mediterranean and Tyrrhenian Sea, respectively. In response to this comment, we will mention the reasons why the AVISO and ROM data time periods are not exactly the same. Also, the consequences of our choice in terms of the results will be stated.

---

## Author Response (AR1)

Dear editor,

A detailed point-by-point response to the reviewers' comments in which the main changes to be performed in the revised version of the manuscript were stated was already uploaded into the system at the Discussion stage. If these files have to be uploaded again, please let us know.

Best regards,
Alba de la Vara on behalf of the co-authors

---

## Author Response (AR2)

**Response to Reviewer 1:**

**This is my second review of the manuscript, which I find slightly improved with respect to the first version. I still find points that I would like authors to consider.**

Thank you for your positive, minor comments. They helped us to attain a more robust and complete version of the manuscript. The responses to the specific comments can be found below.

**As for my comment on the validation and criticism of Figure 2 in the previous review (" From Figure 2, it is difficult to conclude that both patterns agree satisfactorily, partially because of the poor resolution of the figure. Rather, the agreement seems to be only moderate if not arguable"), I still maintain the same objections. In particular, about the poor quality of the Figure. Velocity vectors are not distinguishable, and where they are, its size indicates a sluggish –and unrealistic- surface flow no greater than 1 or 2 cm/s according to the bottom-right reference vector.**

We would like to highlight that the issue posed by the reviewer is mainly due to a visualization problem. For that reason, for the revised version of the manuscript, a new version of Figure 2 has been created (Figure 1R). In this case we set the bottom-right reference vector to 8 cm/s (instead of 5 cm/s, as it was previously). We use this scale as a compromise between the AVISO and ROM grid resolution. In ROM, the resolution decreases from about 7 km near Gibraltar (comparable to AVISO) to about 20 in the Eastern Mediterranean. In the new figure it can be seen that ROM is able to simulate the main features of the ocean circulation in the Western Mediterranean (our area of study), especially the mesoscale activity in the Alboran Sea, with a good representation of the main features of the circulation in the Tyrrhenian Sea.

**It is also the rare practice of using different spatial resolution and time-averaged periods to compare AVISO and ROM data. The new sentence included in the caption neither helps, rather it puts more grain in the mill: if authors have compared the same period, why don't they show the coincidental period instead of two different?**

In our former response to a similar comment that arose in the previous review we showed a series of figures in which it was clear that the differences between using slightly different time periods or exclusively considering the coincidental time period between AVISO and ROM for the validation were minimal and did not have an impact on the extracted conclusions. However, following the reviewer's request, we have generated a new version of Figure 2 for the revised version of the manuscript. This new figure (Figure 1R) only includes the time period which coincides in both AVISO and ROM (1993-2005) and, as stated above, to improve the visualization of the velocity field, the velocity of the reference vector found to the bottom right of the figure has been set to 8 cm/s.

In addition to this, for clarity, in the introductory paragraph of the Results Section of the revised version of the manuscript we state the following: "For estimating ROM present-day climate we use the last 30 years of the historical run, namely from 1976 to 2005, which is widely used as reference period for present climate, then we assess circulation changes by comparing this reference period with the future one (2070-2099). However, for the validation we use the 1993-2005, as this period is coincidental with the AVISO satellite altimetry gridded product". Furthermore, in Section 3.1, in which the Mediterranean Sea circulation is validated, the subsequent sentence has been added: "We remark that the time period chosen for this validation is made to match with AVISO data availability (1993-2005)". Finally, the parts of the text where we mentioned that the mismatch between the time periods used in AVISO and ROM was due to the lack of AVISO data until 1993 have been correspondingly removed to be consistent with the new version of Figure 2.

[Figure]

*Figure 1R: Winter (left column) and summer (right column) averages of present-day geostrophic circulation (vectors, cm/s) and sea-surface height (SSH, colors, cm) from AVISO (A, B) and ROM (C, D). For comparison we use the 1993-2005 period, when both model and observed data are available.*

**The positive contribution of the figure is the similarity of the patterns of large spatial-scale circulation, but the comparison is not so good for the mesoscale. All in all, Figure 2 is of little help for validation. Quite probably, ROM results are better than what can be inferred**
**from the Figure and, therefore, my suggestion is to remove it and show the results of the good correlation found between SSH from ROM and AVISO mentioned (but not shown in the current version) in lines 172-173 instead.**

As it can be appreciated in Figure 1R (Figure 2 of the revised version of the manuscript), ROM provides quite a realistic representation of the basin-scale circulation and its seasonal variability compared to AVISO. Major gyres are also well represented, but the reproduction of mesoscale-size structures is limited by ROM horizontal resolution, which is fair in the western basin (7 km in the Alboran Sea) but relatively low in the Eastern basin (25 km), as shown in Fig. 1 of the manuscript. ROM ocean resolution shows up in the mesoscale representation: mesoscale-size structures are properly reproduced in the Western basin and in the Adriatic Sea, and not so good in the Eastern basin. In short, ROM captures the most prominent dynamical structures depicted from AVISO data, but not all the secondary ones. This was also the case in previous works addressing the Tyrrhenian surface circulation (e.g., de la Vara et al., 2019). The differences between the representation of the dynamical structures in ROM and AVISO is, as explained above, due to the lower resolution of the ocean model relative to AVISO altimeter data. However, this should not be problematic given that the main aim of this work is to study the changes induced in the Tyrrhenian circulation in response to climate change. Despite those differences, good correlation is found between the SSH from ROM and AVISO in both seasons (0.73 for winter and 0.67 for summer).

In response to this comment, the information presented in the above paragraph has been incorporated in Section 3.1 of the revised version of the manuscript. We would like to highlight that the seasonal values of the correlation coefficients for ROM and AVISO SSH taking into account the entire Mediterranean Sea for the 1993-2005 time period have been included in the text, following the reviewer's suggestion (see Table 1R). Since Figure 2, as mentioned above, has been improved and the reasons for the differences in the quality of the representation of mesoscale structuctures has been explained in the revised version of the manuscript, for clarity and consistency with Figure 3, we prefer to keep this figure in the manuscript.

| | | Correlation coefficients |
|---|---|---|
| **AVISO-ROM** **(1993-2005)** | DJF | 0.73 |
| | JJA | 0.67 |
| **AVISO-ROM** **(1976-2005)** | DJF | 0.71 |
| | JJA | 0.66 |

*Table 1R. Winter and summer correlation coefficients found between the SSH from AVISO and ROM for the time periods specified. In the upper row, data for both AVISO and ROM extend from 1993 to 2005. In the lower row, AVISO data expands from 1993 to 2005 and ROM data from 1976 to 2005.*

**The other point in my previous review concerned the forecasted freshening of the Atlantic inflow through Gibraltar. A clarifying sentence is now mentioned in section 3.3, which is OK. However, since this freshening (which is central to the augmented stratification in the Tyrrhenian Sea) is a result imported from previous works, the suitable place for that sentence/comment seems to be the model setup section, where water exchange through the straits of Gibraltar and Dardanelles is mentioned. Obviously, some comments can be maintained in section 3.1, but readers will be grateful for being informed in advance.**

Thank you for your constructive comment. In the revised version of the manuscript, in the Model Setup, where MPIOM, the ocean component of ROM, is described, the following is stated: "As the atmospheric domain is large and encompasses most of the North Atlantic, MPI-ESM influences the oceanic properties through the large-scale forcing on the atmospheric component of ROM. In turn, the large-scale North Atlantic Ocean climate change signal can propagate into the Mediterranean through the open Strait of Gibraltar. This is relevant as it allows accounting for the surface freshening signal projected by the driving MPI-ESM in the eastern North-Atlantic at the end of the 21$^{st}$ century (Parras-Berrocal et al., 2020)".

**The fourth and last conclusion of the manuscript relates to this issue. It is said that water flowing through the Corsica Channel "presents a stronger stratification than at present due to a generalized warming with a saltening of intermediate waters" and the consequences the stratification could possibly have in the formation of deep water in the Gulf of Lions. However, the inclusion of a map with differences in Figures 6 and 8 (good idea showing that map!!) indicates that most likely the main contributor to the future enhanced stratification is the reduced surface salinity. No mention to this fact is made in the conclusion, and it should be done. Hence, the relevance of explaining the source of the low surface salinity of the Atlantic inflow in the introduction section.**

We agree that these are indeed good contributions to the manuscript. Following the reviewer's advice, in the last bullet of the Conclusions Section we now state: "In the future, the northward water

transport across the Corsica Channel towards the Liguro-Provençal basin becomes smaller than today. Also, water that flows through this channel presents a stronger stratification than at present. The reason for this is twofold. On the one hand, a generalized freshening of the Atlantic waters inflowing through Gibraltar causes a reduction of the Mediterranean sea-surface salinity (e.g., Parras-Berrocal et al., 2020). On the other hand, a warming with a saltening of intermediate waters. These changes potentially contribute to the interruption of deep water formation in the Gulf of Lions by the end of the century (Parras-Berrocal et al. 2021)".

To take into account the second part of the comment, in the Introduction Section of the revised manuscript we now the state: "A great advantage of the ROM configuration used in this study is that it represents an open Strait of Gibraltar (see Fig. 1) allowing the propagation of Atlantic Ocean climate change signals into the Mediterranean Sea (Parras-Berrocal et al., 2020; 2021). This is of uttermost importance as the surface freshening of the eastern North Atlantic, which is a robust feature within CMIP5 RCP8.5 projections (Levang and Schmitt, 2020; Soto-Navarro et al.; 2020), may have profound impacts on the Mediterranean Sea (Parras-Berrocal et al., 2020; 2021) in general and on the Tyrrhenian Sea in particular".

**References:**

de la Vara, A., Galan del Sastre, P., Arsouze, T., Gallardo, C., and Gaertner, M.A.: Role of atmospheric resolution in the long-term seasonal variability of the Tyrrhenian Sea circulation from a set of ocean hindcast simulations (1997–2008), Oce. Mod., 134, 51-67, doi:10.1016/j.ocemod.2019.01.004, 2019.

Levang, S.J., Schmitt, R.W.: What Causes the AMOC to Weaken in CMIP5? J. Clim., 33 (4), 1535-1545, doi: 10.1175/JCLI-D-19-0547.1, 2020.

Parras-Berrocal, I., Vazquez, R., Cabos, W., Sein, D., Mañanes, R, Perez-Sanz, J., and Izquierdo, A.: The climate change signal in the Mediterranean Sea in a regionally coupled ocean-atmosphere model, Ocean Sci., doi: 10.5194/os-2019-42, 2020.

Parras-Berrocal, I., Vazquez, R., Cabos, W., Sein, D., Alvarez, O., Bruno, M., and Izquierdo, A.: Will deep water formation collapse in the North Western Mediterranean Sea by the end of the 21st century?, Earth and Space Science Open Archive, doi:10.1002/essoar.10507698.1, 2021.

Soto-Navarro, J., Jordá, G., Amores, A., Cabos, W., Somot, S., Sevault, F., et al.: Evolution of Mediterranean Sea water properties under climate change scenarios in the Med-CORDEX ensemble, Clim. Dyn., 54(3), 2135-2165, doi:10.1007/s00382-019-05105-4, 2020.

**Response to Reviewer 2:**

**This is my second review of this manuscript. My main concerns with the first version were related to the model description, its validation and the explanation of the reduction in the sea-surface salinity (SSS) described in the                                                                                               results and partially responsible of the increase of the stratification in the Tyrrhenian basin. In their response, the authors have addressed most of these points satisfactorily, although I still have a couple suggestions that in my view could help clarify these aspects of the paper. Therefore, I recommend the publication of the article after a minor revision of these points.**

Thank you for your positive, minor comments. They helped us to attain a more robust and complete version of the manuscript.

**Regarding the model validation, in their response to my previous comments, the authors show that the model currents and SSH are well correlated with AVISO, and also with the CMENS reanalysis product, especially in the winter season (using the spatial correlation). They also show that ROM model improves the SSS estimation with respect to MPI-ESM–LR for the whole Mediterranean, and performs a relatively good representation of the SSS salinity seasonal cycle and interannual variability. However, they do not include any of these results in the revised version of the MS. I understand that including the new figures and describing the results is not necessary and would increase the extent of the manuscript, but it could be included in the supplementary information, with a short summary of the main results in the main text. That would give the reader a better idea of the model accuracy.**

We agree that this information could be interesting for the reader. As to the correlation coefficients between the Mediterranean SSH from ROM and AVISO, these have been included in the last paragraph of Section 3.1 (see Table 1R). In addition to this, by the end of the Model Setup Section we now state that this ROM configuration has been extensively validated in Parras-berrocal et al. (2020), but to ease the reading we provide in Supplementary Material the validation of seasonal and interannual SSS variability. The figures included in the Supplementary Material have been briefly described.

|  |  | Correlation coefficients |
|---|---|---|
| **AVISO-ROM**

**(1993-2005)** | DJF | 0.73 |
|  | JJA | 0.67 |
| **AVISO-ROM**

**(1976-2005)** | DJF | 0.71 |
|  | JJA | 0.66 |

*Table 1R. Winter and summer correlation coefficients found between the SSH from AVISO and ROM for the time periods specified. In the upper row, data for both AVISO and ROM extend from 1993 to 2005. In the lower row, AVISO data expands from 1993 to 2005 and ROM data from 1976 to 2005.*

**With respect to the freshening of the Atlantic waters inflowing through Gibraltar. The authors justification is that it is also reported by Parras-Berrocal et al., (2020, 2021) and Soto-Navarro et al. (2020). In the case of the works of Parras-Berrocal et al., they analyze the same simulation studied here so of course they find the same results. Soto-Navarro et al. (2020) studied an ensemble of climatic simulations (historical and 21st century projections) for the Mediterranean Sea, and, indeed, they find that some of them project is a decrease in the SSS in the Western Mediterranean in the future. These authors hypothesize that this could be caused by the freshening of the North Atlantic surface waters as a consequence of the ice melting in the Arctic. Is this the hypothesis assumed here? I miss a couple of sentences in the new version of the article regarding the physical interpretation of this important result. In addition, are there other papers in the literature showing similar results, i. e., a reduction of the SSS in the Northeastern Atlantic by the end of the 21st century? Which are the explanations given in these previous works? I think that a short paragraph in the results or discussion section summarizing previous results about this point would be very interesting and would clarify one of the main results of the article.**

Thank you for your comment. To account for it, we have introduced some changes in Section 4.2 in the revised version of the manuscript so that now the information is clearly presented. In particular, we state the following: "The decrease in surface salinity in the subpolar North Atlantic and in the eastern limb of the North Atlantic subtropical gyre is a robust feature of CMIP5 multimodel ensemble (Levang and Schmitt, 2020; Soto-Navarro et al., 2020) and the establishment of its causes is matter of intense research. Very recently Sathyanarayan et al. (2021) pointed out that besides changes in surface freshwater fluxes, changes in salinity in the Atlantic may be related to changes in wind-stress and circulation, which in turn are related to changes in surface warming. Finally, they remark that the projected AMOC weakening may play an important role".

**References:**

Levang, S.J., Schmitt, R.W.: What Causes the AMOC to Weaken in CMIP5? J. Clim., 33 (4), 1535-1545, doi: 10.1175/JCLI-D-19-0547.1, 2020.

Parras-Berrocal, I., Vazquez, R., Cabos, W., Sein, D., Mañanes, R, Perez-Sanz, J., and Izquierdo, A.: The climate change signal in the Mediterranean Sea in a regionally coupled ocean-atmosphere model, Ocean Sci., doi:10.5194/os-2019-42, 2020.

Parras-Berrocal, I., Vazquez, R., Cabos, W., Sein, D., Alvarez, O., Bruno, M., and Izquierdo, A.: Will deep water formation collapse in the North Western Mediterranean Sea by the end of the 21st century?, Earth and Space Science Open Archive, doi:10.1002/essoar.10507698.1, 2021.

Sathyanarayanan, A., Köhl, A., Stammer, D.: Ocean Salinity Changes in the Global Ocean under Global Warming Conditions. Part I: Mechanisms in a Strong Warming Scenario, J. Clim., 34(20), 8219-8236. Retrieved Nov 9, 2021, from https://journals.ametsoc.org/view/journals/clim/34/20/JCLI-D-20-0865.1.xml, 2021.

Soto-Navarro, J., Jordá, G., Amores, A., Cabos, W., Somot, S., Sevault, F., et al.: Evolution of Mediterranean Sea water properties under climate change scenarios in the Med-CORDEX ensemble, Clim. Dyn., 54(3), 2135-2165, doi:10.1007/s00382-019-05105-4, 2020.